# Can We Generate Realistic Hands Only Using Convolution?

## Abstract

Despite their extensive use in generating hyperrealistic images, image generative models are prone to generating abnormal details and malformed features. One of the most prominent examples of this phenomenon is synthesising contorted and mutated hands and fingers. Convolution serves as the backbone of many state-of-the-art image generative models, all of which are subject to the aforementioned phenomenon. We investigate whether adding a single channel, comprising horizontal and vertical coordinate information, to the input channels of convolution layers can alleviate this issue. We show that the answer is "yes"! We demonstrate this, in a GPU-poor setup, on two families of generative models, Generative Adversarial Networks (GANs) and Variational AutoEncoders (VAEs) trained on the Hand Gesture dataset. The hand images generated by models employing our method surpass those of models using simple convolution by a significant margin. We further validate the results for generating human faces using models trained on the CelebA-HQ dataset, demonstrating our models consistently yield superior images compared to those generated using simple convolution.

## 1 Introduction

*Convolutional Neural Networks* have played an important role in machine learning and computer vision since early 1980s (Fukushima & Miyake, 1982) and especially after they were successfully used for reading handwritten digits (LeCun et al., 1989). They have since been deployed ubiquitously to reach super-human accuracy in image classification and object detection (Simonyan & Zisserman, 2015; He et al., 2016), and have been used in other areas of machine learning such as in families of Image Generative models as in *Generative Adversarial Networks* (*GAN*) (Goodfellow et al., 2014; Radford et al., 2016; Karras et al., 2018) and *Variational Autoencoders* (*VAE*) (Kingma & Welling, 2014; Ramesh et al., 2021; 2022), etc.

Researchers have also considered alternatives to CNN such as *Vision Transformers* (*ViT*) (Dosovitskiy et al., 2021; Parmar et al., 2019). ViT was motivated by the successful adoption of attention mechanism (Bahdanau et al., 2015) and transformers (Vaswani et al., 2017) in natural language processing, and has achieved great success in computer vision tasks. ViT are generally considered to be better than CNN. However, recent research has shown that under fair experimental settings, CNN can even surpass ViT in terms of robustness and expressiveness (Pinto et al., 2021; Bai et al., 2021; Pinto et al., 2022).

One of the common criticisms of CNN is that they differ from human perception in many ways, and therefore, do not always perform as intended (Kosiorek et al., 2019). This is sometimes attributed to the small size of their receptive field, which prevents them from capturing geometric wide-apart information from the images (Luo et al., 2016; Kosiorek et al., 2019; Dai et al., 2017). There is already a body of research on how to address this problem. Some of the most notable approaches are using CNN alongside transformers (Hendria et al., 2021), using deformable CNN (Dai et al., 2017), and CoordConv (Liu et al., 2018).

To address the deficiencies of state-of-the-art image generative models, inspired by Liu et al. (2018), we introduce an enhancement on their approach and introduce a technique for improving the performance of CNN-based image generative models. We call this technique *Geomety-Aware Convolution* (*GeoConv*). GeoConv improves CNN's performance by allowing them to capture and encode the geometric information from the images, instead of encoding only local information limited by the

convolution's filter size. This is achieved via providing the convolution operation the geometric information about the images by concatenating the *Geometry Channel* to the input of the convolution. We demonstrate GeoConvs' ability to encode geometric information about the images in the following applications:

- Generative Adversarial Networks for Face and American Sign Language Hand Gesture generation

- Variational AutoeEcoders for Face and American Sign Language Hand Gesture generation

The layout of this paper is as follows. We first discuss the related work in the rest of this section. In Section 2, we lay out the details about GeoConv's architecture, and prove the technical results. In Section 3, we evaluate GeoConv's performance by comparing it against simple convolution and CoordConv (Liu et al., 2018).

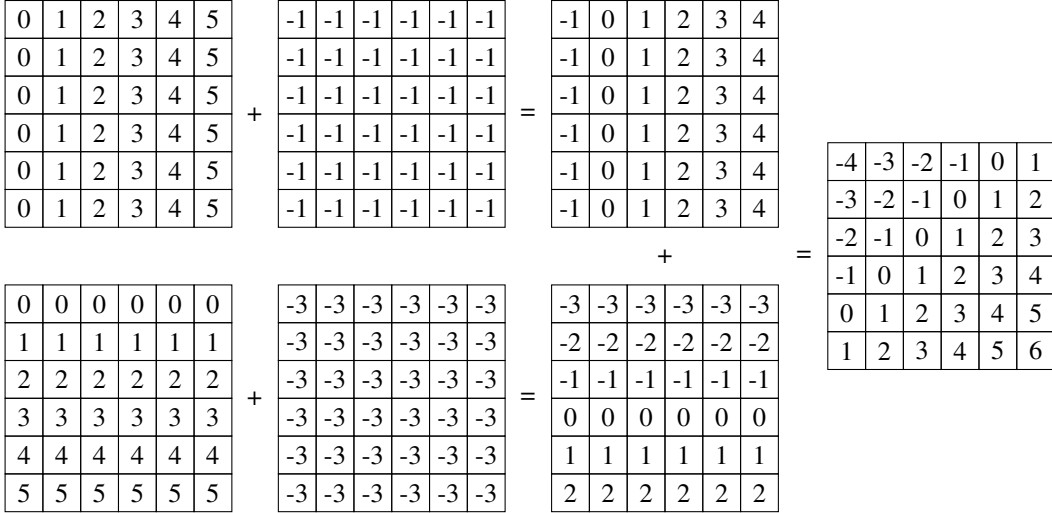

Figure 1: A demonstration of how 2D $6 \times 6$ geometry channel in GeoConv is created. In the implementation, the random horizontal and vertical shifts are combined into a single random number, which defines their summation..

## 1.1 RELATED WORK

Liu et al. (2018) demonstrated the failure of CNN in transforming the spatial representation between the input and output. They introduce CoordConv to address this problem of CNN. *CoordConv* adds two channels to the convolution's input, called coordinate channels.

As demonstrated in Liu et al. (2018), the addition of the coordinate channels improves the performance of CNN in supervised coordinate task, image classification, object detection, GAN and reinforcement learning. However, all of the examples considered in Liu et al. (2018) are quite different from real-world applications of CNN.

Nevertheless, CoordConv has gained traction in an array of applications since introduction (Wang et al., 2019; Long et al., 2020; Choi et al., 2020; Lee & Kim, 2019).

Recently, researchers have applied similar ideas to attention networks. Xie et al. (2022) have used *deep coordinate attention networks* for obtaining super resolution from single images, and Hou et al. used a similar approach for designing mobile attention networks (Hou et al., 2021). However, both of these approaches are fundamentally different from CoordConv and the work presented here, not only because they are considering attention mechanism instead of convolution, but mainly because the use of "coordinate" in those works refers to capturing the positional information in the channels by aggregating the spatial information alongside the vertical and horizontal direction.

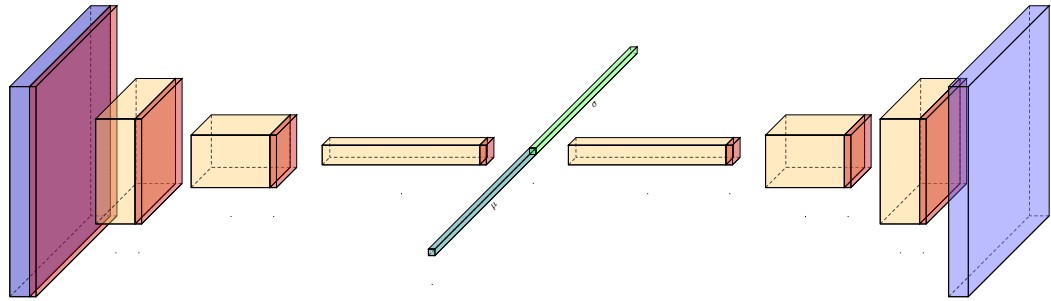

Figure 2: GeoConv in a VAE. The yellow blocks indicate the channels resulting from the previous layer's convolution operations, and the orange channels indicate the geometry channels appended to them during the GeoConv's operation before applying the next convolution operation.

## 2 GEOMETRY-AWARE CONVOLUTIONAL NEURAL NETWORKS

CoordConv addresses the inability of CNN to capture the positional information of images by adding two coordinate channels, one for each dimension, before applying the convolution operation. These channels are similar to the two leftmost channels in Figure 1. These two channels can come from other coordinate representations such as polar coordinate as well.

This considerably improves the performance of CNN in a range of regression tasks (Wang et al., 2019; Long et al., 2020; Choi et al., 2020; Lee & Kim, 2019).

However, as we show in this paper, there are two problems with CoordConv.

1. CoordConv can learn bias or unsought features from the dataset, which can lead to poor performance.
2. CoordConv operation is not optimal.

Here, we introduce the *Geometry-aware Convolutions* or *GeoConv* for short, which not only resolves the inability of CNN in capturing positional information, but also addresses the aforementioned issues with CoordConv.

**Solution to Problem 1.**   The problem with adding these channels to the images is that in addition to learning the spatial information about the image content, the model can also develop correlation between a feature and where it appears in the image rather than their relation (or location) with respect to other features in the image. For instance, if due to the bias in the training dataset a feature mostly appears in the middle of the images, the model begins to develop bias for the position of that feature. We demonstrate this in an example in the Appendix in the supplementary materials. Developing such correlations is undesirable in most real world scenarios. For example, when training face recognition models, the images are nicely cropped and the faces are centred in the training set; however, in the real world, where the model is deployed, this is rarely the case.

We, therefore, introduce random shifts to coordinate channels to prevent the model from learning unwanted positional bias. The random shifts are demonstrated in the second column of Figure 1. Note that as illustrated in the third column of Figure 1, the effect of these random shifts is slightly different from the shifts commonly applied in data augmentation as the values of the pixels on the edge are defined similarly to the ones in the centre.

**Solution to Problem 2.**   Adding two coordinate channels to the images is unnecessary and inefficient since the filters corresponding to these channels do not use their full potential. This is formally stated and proved in Theorem 1.

**Theorem 1.** *A $k_1 \times k_2$ convolution filter on the horizontal (vertical) coordinate channel in Coord-Conv does not extract any more information than a $1 \times k_2$ ($k_1 \times 1$) convolution filter.*

*Proof.* Since the proof for the vertical coordinate channel is similar to the proof for the horizontal coordinate channel, we only prove this for the horizontal coordinate channel. Let $f = (f_{i,j})$ be

the convolution filter corresponding the horizontal coordinate channel $h$ in CoordConv, and for $1 \leq j \leq k_2$, let $\bar{f}_j = \sum_i f_{i,j}$. At each step of the convolution operation, we have that

$$\sum_i \sum_j f_{i,j} h_{s+i,t+j} = \sum_i \sum_j f_{i,j} h_{s,t+j} = \sum_j (\sum_i f_{i,j}) h_{s,t+j} = \sum_j \bar{f}_j h_{s,t+j}.$$

Hence, we conclude that the $k_1 \times k_2$ filter $f$ does not extract anymore information from the horizontal coordinate channel $h$ than the $1 \times k_2$ filter $\bar{f} = (\bar{f}_j)$. □

Therefore, we instead combine the two coordinate channels into one by adding them together, as illustrated in the rightmost column of Figure 1. The Geometry Channel is then concatenated to the input channels as demonstrated in Figure 2. By using one Geometry channel instead of the two coordinate channels in CoordConv, alongside the random shifts, we achieve superior performance compared to CoordConv using one less filter per convolution, and consequently, $k_1 k_2$ less learnable parameters. This provides us with a model that is easier to train, faster, smaller, and thus, deployable in a wider range of edge devices.

## 3 Evaluation

In this section, we embark on a comprehensive assessment of various conditional and unconditional Generative Adversarial Networks as well as conditional Variational Autoencoders, with a particular emphasis on the integration of Geometry-Aware convolution versus conventional convolution techniques. Our evaluation encompasses both facial and hand gesture image generation tasks, providing insights into the efficacy of these models.

Furthermore, in the pursuit of validating our earlier theorem, we extend our assessments to the domain of calculating the center of mass for a set of points within a 2D plane. To achieve this, we deploy CNNs employing different convolution methodologies, including GeoConv, CoordConv, and standard convolution. This supplementary experiment serves as a compelling testament to Geo-Conv's remarkable proficiency in capturing intricate geometric relationships, both at local and global scales, within images. For those interested in delving deeper into this experiment, we invite you to explore the appendix. Nonetheless, our primary focus in this section remains firmly rooted in our image generation experiments with VAEs and GANs, which constitute the core of our research.

Aside from the mass center experiment, we've included two additional experiments in the appendix due to space limitations. The first experiment focuses on calculating positional bias, demonstrating our method's superiority over simple convolution and Coordconv. The final experiment in the appendix employs conditional VAEs to generate hand gestures, highlighting another instance where geometry-aware convolution enhances image generation quality in this category of generative models.

The code for all the experiments is provided in the supplementary materials. All of the experiments in this section have been performed on one GeForce RTX 4090 GPU and 128 GB of RAM.

### 3.1 Generative Adversarial Networks

GANs Goodfellow et al. (2014) are deployed in an array of vision tasks such as generating realistic photographs of human faces Karras et al. (2018), super resolution Ledig et al. (2017), photo blending Wu et al. (2019), etc. Similarly to VAEs, the performance of convolution depends on the information that the convolutions can obtain both locally and globally from the images.

In this subsection, we present our GAN experiments as follows: First, we discuss DCGAN(Radford et al., 2016) experiments for generating human faces with the CelebA_hq(Karras et al., 2018) dataset. Next, we outline WGAN-GP(Gulrajani et al., 2017) experiments on the same task using CelebA_hq to validate earlier results. Finally, we cover our Conditional WGAN-GP experiments for generating hand images using ASL Hand Gesture dataset (Barczak et al., 2011).

### 3.1.1 DCGAN For Generating Human Faces Using CelebA_hq Dataset

To begin our GAN experiments, we employed a Deep Convolutional GAN (DCGAN) with a generator featuring 11 convolutional and transposed convolutional layers, and a discriminator with

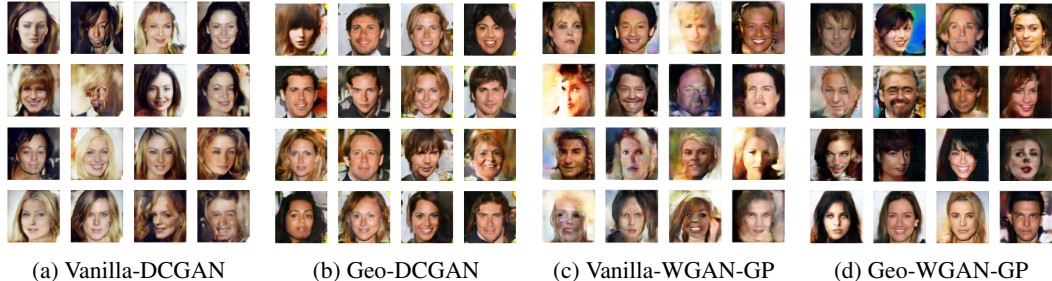

(a) Vanilla-DCGAN      (b) Geo-DCGAN      (c) Vanilla-WGAN-GP      (d) Geo-WGAN-GP

Figure 3: The best performance achieved by different GANs.

10 convolutional layers. These architectures adhere to standard DCGAN practices. We adjusted the network size and parameters to optimize performance for generating human images using the CelebA_hq dataset.

We conducted a 750-epoch training with two models: Geo-DCGAN, incorporating Geometry-Aware Convolutional and Transposed Convolutional layers in both the Discriminator and Generator architectures, and Vanilla-DCGAN, which uses simple Convolution. Besides these layers, all other parameters and architectural components remain consistent between both models. We utilized a 128-dimensional latent space for the Generator. Additionally, we sampled 16 random seeds from this space and visualized the Generator's output for each seed after every training epoch.

The quality of images produced by Vanilla-DCGAN shows a steady improvement during the initial 250 training epochs but declines afterward, as we start to observe mode collapse. In contrast, Geo-DCGAN consistently enhances image quality throughout training, outperforming Vanilla-DCGAN at every epoch since the beginning. Importantly, while Geo-DCGAN faithfully captures the dataset's diversity, Vanilla-DCGAN struggles to do so.

In Figures 3a and 3b, filtering out images with distinctive features leaves Vanilla-DCGAN with only 14 remaining images, of which 2 are male (14.28%) and 12 are female (85.72%). In contrast, Geo-DCGAN generates 16 images, with 6 males (37.5%) and 10 females (62.5%). Notably, the CelebA_HQ dataset comprises 11,057 (36.85%) male and 18,943 (63.15%) female images (Na, 2021), mirroring the distribution effectively captured by Geo-DCGAN.

### 3.1.2 WGAN-GP For Generating Human Faces Using CelebA_hq Dataset

One common challenge encountered in Generative Adversarial Networks (GANs) is known as Mode Collapse, where the model predominantly produces a limited set of identical or very similar outputs, failing to capture the diversity present in the training dataset. To address this issue, various strategies have been proposed in the GAN literature, with Wasserstein GANs (WGANs) (Arjovsky et al., 2017) emerging as a prominent solution.

A prevalent technique employed in WGANs to stabilize training and enhance the quality of generated images involves penalizing the gradient's norm concerning its input (Gulrajani et al., 2017). The Wasserstein GANs utilizing this approach are commonly called WGAN-GP and we use this name in our paper as well. It is crucial to emphasize that while this approach effectively mitigates model collapse, it may introduce a potential trade-off, as it will lead to longer training times due to the computation of second-order gradients and potentially lower image quality.

In our study comparing GANs using Geometry-Aware convolution to conventional GANs with those using simple convolution, we next employed the WGAN-GP architecture to address mode collapse, a limitation observed during extended training of Vanilla-DCGAN. We examined two WGAN-GP models: one, called Vanilla WGAN-GP, used standard convolutional and transposed convolutional layers in both discriminator and generator, while the other utilized Geometry-Aware Convolutional and Transposed Convolutional layers. All other hyperparameters and network components were consistent between these two models.

Each training epoch of WGAN-GP takes about 6 times longer in comparison to the DCGAN due to the computation of second-order gradients in its architecture. To this end, we trained both models for

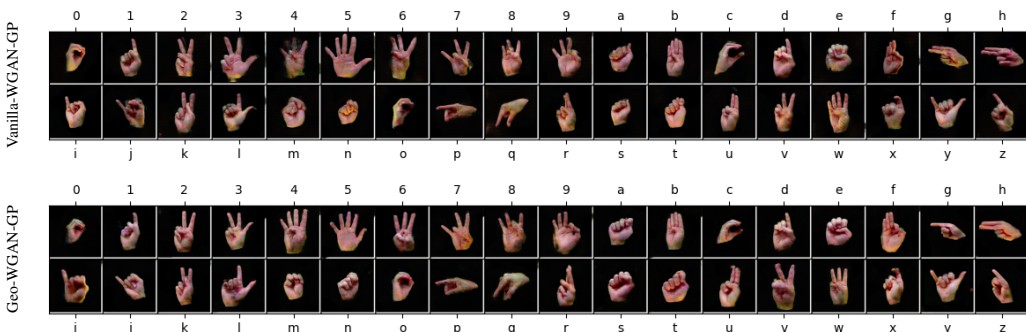

Figure 4: The best performance of Conditional Vanilla-WGAN-GP and Geo-WGAN-GP

125 epochs. Similar to the last experiment, we kept the latent dimension at 128. Figure **??** presents the best performance of each of these two models during the training process.

Throughout the training process, the Geo-WGAN-GP consistently demonstrated significantly superior image generation quality compared to the Vanilla-WGAN-GP at each epoch. While the utilization of WGAN-GP, as opposed to DCGAN, effectively mitigates mode collapse, it presents challenges for both models, particularly the one employing vanilla convolutional layers, resulting in the production of incomplete and less visually appealing images.

Furthermore, with regards to capturing the diversity present in the dataset, the Geo-WGAN-GP proved more adept at generating diverse images, yielding approximately 5 or 6 resembling male characters and 10 or 11 resembling female characters. In contrast, the images generated by the Vanilla-WGAN-GP exhibited only 2 or 3 resemblances to male characters.

### 3.1.3 CONDITIONAL WGAN-GP FOR GENERATING REALISTIC HANDS USING HAND GESTURE DATASET

The Hand Gesture dataset (Barczak et al., 2011) comprises 2524 images depicting American Sign Language (ASL) gestures. This dataset is meticulously annotated, with labels encompassing the entire English alphabet, from 'a' to 'z,' as well as numerical digits ranging from '0' to '9'.

Based on the initial two experiments in the GAN section, our observations indicate that networks incorporating Geometry-Aware Convolutional and Transposed Convolutional layers exhibit significantly enhanced capabilities in generating higher-quality images. Furthermore, they excel in capturing the inherent diversity of the training images compared to networks utilizing conventional Vanilla Convolutional and Transposed Convolutional layers. These notable advantages inspired our investigation into the performance of such models, with a specific focus on addressing a prevalent issue in state-of-the-art image generative models: the generation of unrealistic hand images characterized by unexpected distortions and mutations.

In pursuit of our objectives, and mindful of our GPU resource constraints, we opted for a compact hand dataset featuring relatively high-resolution images. The gesture dataset proved to be an excellent choice for our specific application. Furthermore, in line with contemporary trends in state-of-the-art image generative models, we extended our approach by implementing a Conditional WGAN-GP. This allowed us to explore the dataset's capability to effectively capture and learn the art of generating images based on specified labels.

In line with our consistent experimental methodology, we examine two networks that share identical architectural frameworks, differing solely in their choice of Convolutional and Transposed Convolutional layers. The first network, called Conditional Vanilla-WGAN-GP, employs conventional Convolutional and Transposed Convolutional layers, while the second network, designated as Conditional Geo-WGAN-GP, utilizes their Geometry-Aware counterparts.

Figures 3c and 3d showcases the peak performance achieved by both models. The Conditional Geo-WGAN-GP excels in accurately representing a wide range of hand gestures. Out of the 36 total labels, the Conditional Geo-WGAN-GP adeptly reconstructs hand gestures for all English al-

phabet letters and numbers. In contrast, the Vanilla counterpart encounters challenges with several numbers ('4', '6', '7') and letters ('a', 'g', 'j', 'l', 'm', 'w'). These challenges include distortions, misplaced thumb or finger positions, and incorrect structuring ('h' instead of 'g', 'n' instead of 'm'). Consequently, the Vanilla model falls short in capturing gesture details for 9 out of 36 labels, a notable contrast to the Geometry-Aware counterpart, which successfully generates every letter of the alphabet as well as numbers.

An intriguing observation arises when comparing models employing Vanilla Convolutional and Transposed Convolutional layers with their Geometry-Aware counterparts. Not only do the former models more frequently generate hands with distorted and mutated features, but they also struggle to capture subtle nuances when compared to their Geometry-Aware counterparts. For instance, consider the letters 'm' and 'n' in American Sign Language (ASL). Both letters are notably similar, differing only in the thumb's placement. In 'm', the thumb rests between the little and ring fingers, while in 'n', it's positioned between the ring and middle fingers. Similarly, in the case of 'h' and 'g,' the sole distinction is that 'h' requires only the index finger to be extended outward, whereas 'g' demands both the index and middle fingers to be in that configuration. Regrettably, the Vanilla version falls short in capturing these distinctions, while the models equipped with Geometry-Aware counterparts successfully recognize and replicate these subtleties present in the dataset.

## 3.2 VARIATIONAL AUTOENCODERS

The GAN results underscore the significant performance advantage achieved by models incorporating Geometry-Aware Convolutional and Transposed Convolutional layers when compared to their counterparts using standard Convolutional and Transposed Convolutional layers. This superiority becomes strikingly evident when assessing the qualitative aspects of the generated images, where models leveraging the Geometry-Aware technique consistently produce higher-quality results.

Conventional GAN assessment techniques like Inception Score (IS) and Fréchet Inception Distance (FID) have notable limitations in adequately gauging image quality, as highlighted by Karras et al. (2019) and Borji (2022). For example, FID can exhibit high bias, especially with sample sizes below 50,000, potentially causing overestimation (Chong & Forsyth, 2020). Similarly, IS has drawbacks, including insensitivity to the prior distribution and hypersensitivity to implementation details and model parameters, as noted by Barratt & Sharma (2018).

In order to broaden the scope of our assessment, enabling a more comprehensive evaluation of models employing Geometry-Aware convolution in contrast to their counterparts using simple Convolution, and to derive more insightful quantitative insights into the impact of incorporating spatial coordination information within the convolutional operation, the subsequent subsections delve into our experiments conducted with Variational Autoencoders (VAEs).

VAEs have achieved tremendous success since they first appeared in the literature (Kingma & Welling, 2014). Soon, they were widely used in a range of applications (Kajino, 2019; Singh & Ogunfunmi, 2022; Kovenko & Bogach, 2020). In particular, during the last few years, they have been used for generating images and art from natural language descriptions in DALL.E (Ramesh et al., 2021) and DALL.E-2 (Ramesh et al., 2022). DALL.E-2 does not even use attention in its decoder and uses convolutional layers instead (Ramesh et al., 2022).

The effectiveness of convolutions in VAE applications relies on analyzing both local and global image features (Liu et al., 2018; Kosiorek et al., 2019). In the following sections, we assess VAE performance in a conditional setting for generating human face images using CelebA dataset (Liu et al., 2015)) and ASL hand gestures using hand gesture dataset (Barczak et al., 2011)). We compare models using Geometry-Aware convolution with those using simple convolution.

In our experiments, we utilize a loss function that comprises both the reconstruction loss and KL divergence loss, each assigned a weight. The reconstruction loss, further subdivided into 1-Binary Cross Entropy, 2-Mean Absolute Error, 3-Mean Squared Error over pixel summation, 4-Multiscale Structural Similarity (Wang et al., 2003) between the reconstruction and original images, and 5-Absolute difference between Sobel Edge maps of the original and reconstructed images, serves to enhance the quality of VAE-generated images. We elaborate on how each of these components contributes to our goal of maximizing image quality in the appendix.

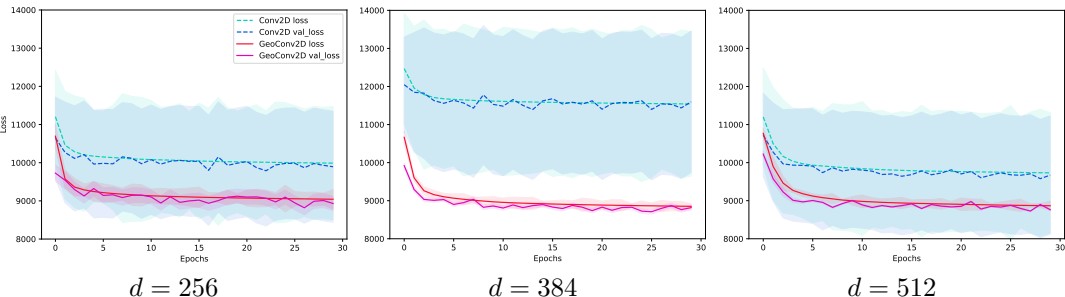

Figure 5: Mean and 95% CI of VAE losses, trained on CelebA for 30 epochs with simple convolution and GeoConv, using latent dimensions $d \in \{256, 384, 512\}$ over five runs (seeds $0, \ldots, 4$).

### 3.2.1 CONDITIONAL VAE FOR GENERATING HUMAN FACES USING CELEBA DATASET

First introduced by Liu et al. (2015) in 2015, the CelebA dataset has grown to become a prominent benchmark in the realm of Computer Vision for predictive and generative applications. Comprising a vast collection of over 202,000 human face images, each annotated with more than 40 binary attributes, it encompasses various features such as hair color, eyeglasses, makeup choices, eyebrow styles, mouth characteristics, and many more. This extensive attribute annotation renders CelebA an ideal dataset for the study of conditional generative models.

In this subsection, we deploy a concise conditional VAE featuring a compact encoder with three convolutional layers and a few dense layers. Likewise, the decoder is streamlined, comprising one dense layer, five Transposed convolutional layers, and one convolutional layer. Our objectives in this subsection are twofold. Firstly, we compare the performance of models using Geometry-Aware convolution to those employing conventional counterparts, scrutinizing the disparities in their generated images. Secondly, we aim for a precise quantitative assessment of this comparison. In each experiment, we trained two VAEs for 30 epochs: one employing conventional Convolutional and Transposed Convolutional layers, and the other using Geometry-Aware counterparts. To ensure robust results, we varied the latent dimensions, testing values of 256, 384, and 512, and conducted training with 5 different seeds, ranging from 0 to 5.

Before assessing the generated images, Figure 5 shows the mean loss and validation loss for each model after 30 training epochs, with the shaded region indicating the 95% Confidence Interval. Across all latent dimensions, models using Geometry-Aware convolution exhibit greater stability and significantly improved performance compared to those using simple Convolution. In latent dimensions 256 and 512, Geometry-Aware VAEs achieve roughly 10% lower losses, while at latent dimension 384, the advantage is even more pronounced, with approximately a 25% reduction in both loss and validation loss.

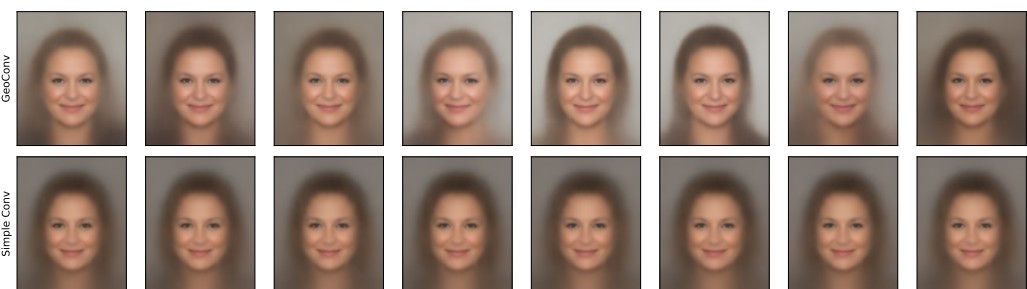

Figure 6: Diversity of images from a conditional-VAE (employing a 384 dimensional latent) decoder after 30 epochs. The first row, by the Geometry-Aware model, has 8 random latent points for one attribute. The second row, from the other model, shares the same attribute and random points.

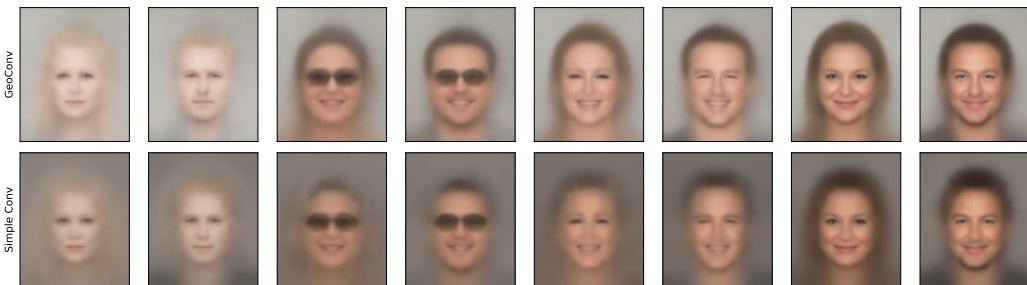

Figure 7: Sharpness of images generated by conditional-VAEs (with a 384 dimensional latent) after 30 epochs. The first row is from the Geometry-Aware model, and the second row is from the other model using simple convolution, both with the same latent point and a set of 8 attributes.

As depicted in Figure 6, the VAE implementing our Geometry-Aware technique demonstrates a notable capacity to produce diverse outputs when presented with various random latent points. In stark contrast, the model employing conventional convolution techniques fails to capture the dataset's inherent diversity, consistently generating similar outputs when given distinct latent points for each set of attributes. For instance, our technique-equipped model exhibits adaptability in attributes like hairstyle, eye and eyebrow styles, and even skin tones, provided such variations align with the attributes specified for the conditional VAE. Conversely, the other model exhibits limited flexibility, yielding less diverse generations.

Furthermore, beyond the diversity of the generated images, as depicted in Figure 7, it is evident that the VAE employing our methodology consistently produces higher-resolution images for different conditions or labels imbued with more pronounced and distinctive features, as compared to the model employing standard convolution techniques.

For a comprehensive view of the attribute sets associated with each column and additional images generated from various latent points, we direct readers to the appendix. Here, we have included only a subset of three generations in this figure, constrained by space limitations.

## 4  CONCLUSIONS AND FUTURE DIRECTIONS

In this study, we introduced Geometry-Aware Convolution, a novel enhancement for Convolutional Neural Networks (CNNs), enabling them to harness two-dimensional coordination information within a single channel. This integration significantly improves the network's ability to capture intricate image features. Our experiments, conducted with Variational Autoencoders (VAEs) and Generative Adversarial Networks (GANs), demonstrated the effectiveness of Geometry-Aware Convolution.

Our results showcased the superiority of models using Geometry-Aware convolution in generating higher-resolution and diverse images across various GAN architectures. Similarly, VAEs employing this technique exhibited substantial improvements in both qualitative and quantitative performance for generating human faces and ASL hand gestures.

Looking ahead, we envision applying this technique to larger-scale commercial generative models, investigating its impact on their performance. Additionally, selectively incorporating this enhancement into specific layers of pre-trained generative models, followed by retraining, shows promise for further performance gains.

Expanding our research to include different families of image generative models, such as those rooted in autoregressive modeling with convolution, presents an exciting avenue for future exploration. Understanding how this technique can enhance these models is a valuable direction for further investigation.

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

# A APPENDIX

## A.1 CALCULATING CENTRE OF MASS EXPERIMENT

GeoConv shows promising advantage compared to simple convolution and Coordconv in tasks that require a geometric understanding of the input. To demonstrate this, we consider the task of calculating the mass centre of points spread out in $28 \times 28$ images, and use a single $3 \times 3$ convolution with ReLU activation as the input layer and a dense output layer with 2 nodes, corresponding to the $x$ and $y$ coordinates of the mass centre.

Table 1: Comparison of average and normalised average performances of GeoConv, CoordConv, and simple CNN.

|  | GeoConv | CoordConv | Simple Conv |
|---|---|---|---|
| Avg. loss | **115.823** | 145.319 | 150.939 |
| Normalised avg. loss | **0.30116** | 0.33227 | 0.36656 |
| # of best performances | **23** | 9 | 17 |

Our motivation for choosing this task and configuration is that it requires the models to have a good understanding of the locations of a varying number of points spread out in the plane with a single convolution layer. Therefore, the models need to obtain a geometric and global knowledge of where the points are, rather than a local knowledge provided by simple convolutions.

To cover different scenarios and have a comprehensive comparison between the architectures, we trained the networks on 7 synthesised datasets, each containing $60,000$ images with point density $d$, where

$$d \in \mathcal{D} = \{0.001, 0.003, 0.01, 0.03, 0.1, 0.3, 0.9\}.$$

The motivation for this choice of $\mathcal{D}$ is to cover different densities of points starting from $0.001$ and increasing geometrically with a factor of roughly 3, up to $0.9$ density.

We then evaluated the performances of each of the networks on 7 test datasets, each containing $10,000$ images with density $d \in \mathcal{D}$. All of the networks were trained by defining the loss function to be the Euclidean distance between the predicted mass centre and the true mass centre. The performances over the test datasets were calculated by averaging the Euclidean distance of the predicted and true mass centres. The results are available in Tables 2-8.

We have summarised the results in Table 1. As demonstrated in Table 1, GeoConv with 23 best performances has the highest number of best performances out of the total 49 experiments, followed by simple convolution with 17 and CoordConv with 9 best performances.

Furthermore, on average, GeoConv is $23.2\%$ and $20.3\%$ more accurate in finding the centre of mass than simple convolution and CoordConv, respectively. To obtain a fairer comparison between the three architectures, we have also provided the normalised average performances in Table 1. GeoConv outperforms simple and CoordConv again by $17.8\%$ and $9.4\%$, respectively.

Table 2: Loss table for models trained on the training dataset with density 0.001.

| Test ratio | 0.001 | 0.003 | 0.01 | 0.03 | 0.1 | 0.3 | 0.9 |
|---|---|---|---|---|---|---|---|
| GeoConv | 2.581 | **3.598** | **18.87** | **79.65** | **296.7** | **916** | **2777** |
| CoordConv | **2.267** | 4.630 | 27.02 | 107.5 | 392.9 | 1208 | 3654 |
| Simple Conv | 2.438 | 4.622 | 24.88 | 100.7 | 370.3 | 1140 | 3449 |

Table 3: Loss table for models trained on the training dataset with density: 0.003.

| Test ratio | 0.001 | 0.003 | 0.01 | 0.03 | 0.1 | 0.3 | 0.9 |
|---|---|---|---|---|---|---|---|
| GeoConv | 5.435 | 2.640 | **2.871** | **6.01** | **20.12** | **66.90** | **211.4** |
| CoordConv | 4.530 | 2.112 | 3.553 | 18.33 | 76.43 | 242.8 | 742.3 |
| Simple Conv | **4.356** | **2.104** | 4.025 | 21.25 | 87.28 | 276.4 | 844.0 |

Table 4: Loss table for models trained on the training dataset with density 0.01.

| Test ratio | 0.001 | 0.003 | 0.01 | 0.03 | 0.1 | 0.3 | 0.9 |
|---|---|---|---|---|---|---|---|
| GeoConv | 6.381 | 3.180 | 1.291 | 4.558 | 24.15 | 80.95 | 251.6 |
| CoordConv | 9.380 | 4.875 | 1.971 | **2.978** | **8.45** | **14.17** | **15.1** |
| Simple Conv | **6.329** | **3.145** | **1.261** | 4.608 | 24.48 | 82.07 | 255.1 |

Table 5: Loss table for models trained on the training dataset with density 0.03.

| Test ratio | 0.001 | 0.003 | 0.01 | 0.03 | 0.1 | 0.3 | 0.9 |
|---|---|---|---|---|---|---|---|
| GeoConv | 9.36 | 5.008 | 2.142 | 1.095 | **1.495** | **4.72** | **13.14** |
| CoordConv | 11.15 | 6.370 | 2.803 | 1.145 | 4.580 | 11.74 | 14.90 |
| Simple Conv | **6.84** | **3.668** | **2.133** | **0.890** | 7.321 | 29.40 | 95.90 |

Table 6: Loss table for models trained on the training dataset with density 0.1.

| Test ratio | 0.001 | 0.003 | 0.01 | 0.03 | 0.1 | 0.3 | 0.9 |
|---|---|---|---|---|---|---|---|
| GeoConv | **7.371** | **3.948** | 2.405 | 1.925 | 0.610 | 5.696 | 23.21 |
| CoordConv | 7.548 | 4.042 | 2.377 | 1.837 | **0.601** | 5.216 | 21.29 |
| Simple Conv | 8.369 | 4.426 | **2.164** | **1.398** | 0.667 | **2.916** | **12.06** |

Table 7: Loss table for models trained on the training dataset with density 0.3.

| Test ratio | 0.001 | 0.003 | 0.01 | 0.03 | 0.1 | 0.3 | 0.9 |
|---|---|---|---|---|---|---|---|
| GeoConv | **6.942** | **3.859** | 2.875 | 2.988 | 2.440 | 0.350 | 7.430 |
| CoordConv | 7.874 | 4.163 | 2.279 | 1.895 | 1.506 | **0.342** | 4.474 |
| Simple Conv | 9.035 | 4.841 | **2.231** | **1.300** | **0.789** | 0.348 | **1.467** |

Table 8: Loss table for models trained on the training dataset with density 0.9.

| Test ratio | 0.001 | 0.003 | 0.01 | 0.03 | 0.1 | 0.3 | 0.9 |
|---|---|---|---|---|---|---|---|
| GeoConv | 9.228 | **5.114** | **2.61** | **1.75** | **1.26** | **0.888** | 0.349 |
| CoordConv | **5.176** | 5.655 | 7.66 | 8.44 | 8.07 | 6.095 | **0.147** |
| Simple Conv | 5.221 | 7.904 | 10.67 | 11.39 | 10.77 | 8.085 | 0.156 |

## A.2 ADDRESSING POSITIONAL DEPENDENCIES

We demonstrate this by considering the task of labelling $64 \times 64$ images that contain Greek numbers I, II. There is exactly one image for each of the numbers in the training set. In the training set, the Greek numbers are centred in the image.

Then, three convolutional models using GeoConv, simple convolution, and CoordConv are created. each of the models have a convolutional layer as their input layer. All of the convolutions use a single $3 \times 3$ filter with $2 \times 2$ strides on the input image. The only other layers, is the output layer, which is a dense layer of size 3. The models are trained on the training set using the categorical cross entropy loss.

The test set also contains $64 \times 64$ images that contain the Greek numbers I, II, and III. However, the test set is the set of all the $6464$ images with different geometric translations of these numbers (e.g. III appears somewhere on the top right of the image instead of the centre). The result of the evaluation is available in Table 9.

As you can see in Table 9, despite having the highest number of learnable parameters, CoordConv has the worst performance amongst all the architectures due to the positional bias learned during the training.

Table 9: The average loss and accuracy of the models when the numbers are moved to all of the possible positions in a $64 \times 64$ canvas

|  | GeoConv | CoordConv | Simple Conv |
|---|---|---|---|
| Loss | 1.229 | 1.890 | **1.170** |
| Accuracy (%) | **36.48** | 33.85 | 36.13 |
| # train param. | 3,093 | 3,102 | 3,048 |

## A.3   VAEs

In Variational AutoEncoders, the choice of loss functions plays a crucial role in determining the quality of the generated images. The combination of various loss functions can help improve the quality of the generated images by encouraging the VAE to produce reconstructions that are closer to the true data distribution. To this end, in both our experiments on VAEs, the one included in the main body and also the one which will be presented in the following subsection, we used a weighted combination of 5 different loss functions in our reconstruction loss and tuned the weights of these loss functions such that both architectures, using Geometry-Aware convolution and simple convolution, are optimized in terms of the quality of the images they generate. Here, we name and break down how each loss function contributes to the overall improvement:

1. **Binary Cross Entropy (BCE) over all pixels:**

   - BCE loss is often used as a pixel-wise reconstruction loss in VAEs when dealing with binary data or data that can be modeled as binary (e.g., images with pixel values in [0, 1]).
   - It encourages the VAE to produce reconstructions that are statistically similar to the input data in a pixel-wise manner.
   - Weighting this loss more heavily would prioritize accurate pixel-level reproduction.

2. **Mean Squared Error (MSE) over all pixels:**

   - MSE is a common choice for reconstruction loss when working with continuous data (e.g., grayscale images with pixel values in [0, 255]).
   - It penalizes larger errors more heavily and can be more sensitive to outliers than BCE.
   - Including this loss helps in reducing pixel-wise differences between the input and reconstructed images.

3. **Mean Absolute Error (MAE) over all pixels:**

   - MAE is another option for continuous data and is less sensitive to outliers than MSE.
   - Like MSE, it helps reduce pixel-wise differences between input and reconstructed images, though the magnitude of errors is emphasized differently.

4. **Multi-scale Structural Similarity (SSIM):**

   - SSIM is a perceptual loss that assesses structural similarity between images, considering luminance, contrast, and structure.
   - Incorporating SSIM encourages the VAE to generate images that are not just pixel-wise accurate but also perceptually similar to the input images.
   - It helps capture high-level features and improves the visual quality of generated images.

5. **Absolute difference of Sobel edge maps:**

   - Sobel edge maps highlight edges and gradients in images, and taking the absolute difference of these maps encourages the VAE to reproduce edges accurately.
   - This loss can help improve the sharpness and structural details in the generated images.

Combining these loss functions in a weighted sum facilitates striking a balance between different aspects of image quality.

### A.3.1 CONDITIONAL VAE FOR GENERATING HAND GESTURE IMAGES USING ASL HAND GESTURE DATASET

The findings from the experiment on VAEs presented in the main body underscore the significant enhancements achieved by incorporating Geometry-Aware Convolution into a VAE. These enhancements manifest in improved qualitative and quantitative performance, as well as a heightened capacity to capture the rich diversity inherent in the dataset.

In alignment with our experiments in the GAN section, we have also conducted a series of experiments for Variational Autoencoders (VAEs) aimed at generating hand gestures. These VAE models were trained using the Hand Gesture dataset to investigate the impact of our convolutional approach on their performance. This investigation holds particular interest due to the stark differences between the Gesture dataset and the previously examined CelebA dataset.

While CelebA boasts a vast and diverse collection, comprising approximately 200,000 human face images, the hand gesture dataset contrasts significantly. It consists of just over 2,500 images, each sharing a similar appearance, primarily differing based on the represented alphabet or number. Consequently, this dataset introduces a distinct set of challenges for the models, demanding a unique set of representations and features to be gleaned from these images.

To this end, we train two conditional VAEs on the training section of the gesture dataset for 100 epochs. It is essential to emphasize that the architectural configuration of these VAEs remains identical to those discussed in the other VAE experiment for human face image generation. The sole distinction lies in the fine-tuning of the parameters specific to the hand dataset. Despite the relatively modest size of this VAE, it assumes significance within the context of a relatively small hand dataset, comprising a mere 2500 images. The specifications and weights assigned to each loss component in the reconstruction loss mirror those employed in our prior experiment. In alignment with our previous experiments, the key disparity between the two trained VAEs in this context emerges from their utilization of distinct Convolutional strategies. One VAE employs conventional Convolutional and Transposed Convolutional layers, while the other adopts our proprietary approach featuring Geometry-Aware Convolutional and Transposed Convolutional layers.

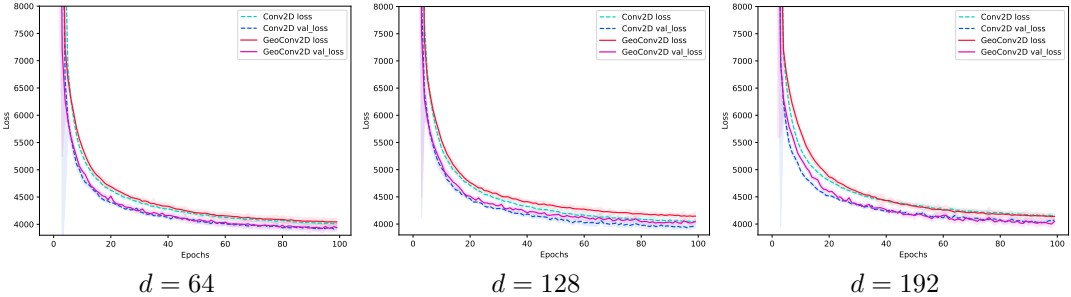

Figure 8: Mean and 95% CI of VAE losses, trained on Hand Gesture for 100 epochs with simple convolution and GeoConv, using latent dimensions $d \in \{64, 128, 192\}$ over five runs (seeds $0, \ldots, 4$).

In line with our approach in the other experiment on VAEs, we conducted experiments employing three distinct latent dimensions: 64, 128, and 192. Furthermore, each model underwent training with five different random seeds, ranging from 0 to 4 to provide us with more reliable data points on the results. The loss trajectories for training and validation of both models throughout 100 epochs of training are visualized in Figure 8. As anticipated, the training and validation loss outcomes exhibit remarkable similarity, with differences consistently falling below half a percentage point across various latent dimensions. This conformity in results aligns with our expectations, considering that the Gesture dataset, unlike the CelebA dataset, is characterized by its small size and limited diversity. Consequently, even relatively straightforward VAE architectures exhibit the capability to capture the primary underlying structures of the dataset with relative ease.

Figure 9 visually presents the generated images produced by both conditional Variational Autoencoders (VAEs). When it comes to faithfully representing the gestures corresponding to their labels, both models have demonstrated commendable performance. Although they may not produce high-

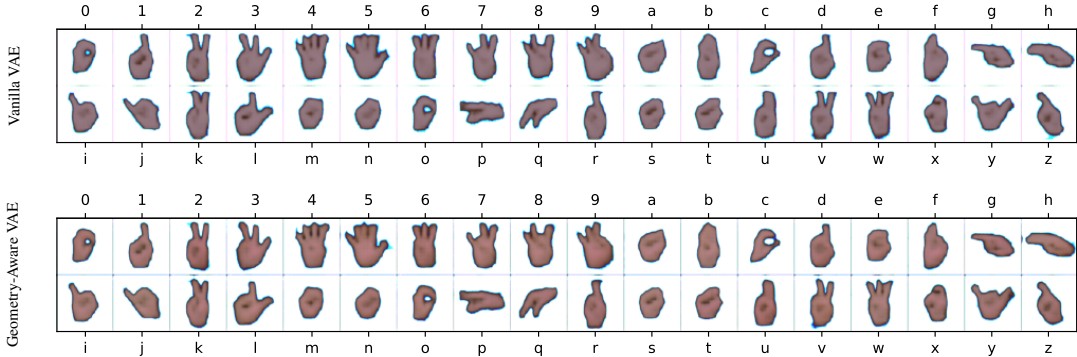

Figure 9: The top row features hand images generated with simple Convolution in the VAE, while the bottom row showcases images generated by the VAE using Geometry-Aware Convolution. Both VAEs share an identical architecture and employ a 192-dimensional latent space.

resolution images akin to Generative Adversarial Networks (GANs), they are indeed generating images that closely resemble the labels they are intended to represent.

Digging deeper into the nuances, it becomes evident that the images generated by the Geometry-Aware VAE exhibit a higher degree of fidelity in terms of capturing the hand colors present in the dataset. These images also manifest a greater degree of realism, steering clear of the uncanny valley effect. Notably, this observation held true consistently across all the experiments conducted with varying latent dimensions.

