# OpenReview forum: "Can We Generate Realistic Hands Using Only Convolution?"
_ICLR.cc/2024/Conference — ICLR 2024 Conference Withdrawn Submission_

### Official Review · Reviewer_NRKz · 2023-10-29

**Soundness:** 2 fair
**Presentation:** 2 fair
**Contribution:** 2 fair
**Rating:** 3
**Confidence:** 5

**Summary:**

The paper focuses on the problem of the phenomenon of synthesizing contorted and mutated hands and fingers. The authors believe that the reason leading to such a phenomenon lies in the so-called “position bias” derived from the convolution operation. To address this, they propose a Geometry-Aware Convolution (GeoConv) that introduces a random shift to prevent the model from learning position bias. The experiments show the proposed GeoConv performs better than traditional convolution in GAN-based and VAE-based models.

**Strengths:**

This paper seeks to investigate the convolution itself to address the phenomenon of “position bias”, which is a more in-depth idea and deserves to be encouraged.

**Weaknesses:**

*Method
1. The concept of the “position bias” is unclear. Does it mean that we should focus on the position of feature w.r.t. other features in the image rather than only considering the static position in the current image?
2. Is the phenomenon of synthesizing contorted and mutated hands and fingers really caused by the “position bias”? Any proof? Besides, does the “position bias” really derive from the intrinsic limitations of convolution? If yes, does it mean the “position bias” is CNN-specific while other architectures like Transformer would not lead to such a phenomenon?
3. The contributions are limited.
a) It would be better to focus on a general generative model (i.e., architecture-independent) instead of a CNN-based model only, which narrows the range of usage for the proposed method.
b) What is the difference between the proposed random shift method and common data augmentation (e.g., random crop), which can also alleviate the position bias?
4. For Theorem 1, it seems hard to claim as a Theorem.

*Experiment

The experiments are not enough, and most of the descriptions are about the details of the existing methods or implementations.

*Overall

I believe that this paper may not be prepared for publication due to its limited contributions, including a basic methodology, inadequate experimental work, and unclear or ambiguous claims and descriptions.

**Questions:**

Please refer to Weaknesses.

---

### Official Review · Reviewer_oSXB · 2023-10-30

**Soundness:** 2 fair
**Presentation:** 1 poor
**Contribution:** 2 fair
**Rating:** 3
**Confidence:** 4

**Summary:**

This paper introduces a technique called Geometry-Aware Convolution (GeoConv) to improve the performance of CNN-based image generative models. Inspired by previous work by Liu et al. (2018), GeoConv aims to address the limitation of capturing only local information by traditional convolution. The technique incorporates geometric information into the convolution operation by adding a Geometry Channel to the input, thus enabling the models to understand broader geometric relations within images. The paper demonstrates the effectiveness of GeoConv through applications in Generative Adversarial Networks and Variational AutoEncoders for generating faces and American Sign Language hand gestures. Through comparative evaluations with simple convolution and CoordConv, the paper shows that GeoConv leads to better performance in capturing and encoding geometric information in images.

**Strengths:**

1. The paper introduces Geometry-Aware Convolution, a novel enhancement for CNNs, which improves their ability to capture intricate image features.

2. Results demonstrate that models using Geometry-Aware convolution generate diverse images across various GAN architectures, as well as showing improvements in the performance of VAEs for generating human faces and ASL hand gestures.

**Weaknesses:**

1. The paper's structure should be reorganized. For instance, the introduction of background information about VAE in Section 3.2 and the introduction of the CelebA dataset in Section 3.2.1 would be better placed in a separate section dedicated to experimental details.

2. On page 4, the authors claim, "we achieve superior performance compared to CoordConv using one less filter per convolution, and consequently, k1k2 less learnable parameters. This provides us with a model that is easier to train, faster, smaller, and thus, deployable in a wider range of edge devices." This statement may not be suitable because the reduction in learnable parameters with GeoConv, compared to CoordConv, is negligible. It's difficult to make a compelling case that such a minor reduction in parameters would have a substantial impact on model training and inference speed, even when considering resource-constrained edge devices.

3. The author's experiments were conducted on low-resolution images (e.g., the results in Figure 3). I understand that this might be due to limited computational resources. However, based solely on the experimental results presented in the current paper, I cannot confirm whether GeoConv remains effective for high-resolution image generation tasks.

4. The paper's writing appears to be somewhat rough, as evidenced by statements like "Figure ?? presents the best performance of each of these two models during the training process" on page 6. It seems that the authors may not have thoroughly proofread their paper before submitting it.

**Questions:**

I believe that this paper needs further improvement, both in terms of experiments and writing, to meet the criteria for acceptance at ICLR. For details, please refer to the "Weaknesses".

---

### Official Review · Reviewer_1G3f · 2023-10-31

**Soundness:** 2 fair
**Presentation:** 2 fair
**Contribution:** 1 poor
**Rating:** 3
**Confidence:** 3

**Summary:**

The paper proposes enhancing convolutional layers in generative models with a hand-crafted geometry channel that encodes positional information. Based on a number of experiments with GANs and VAEs on CelebA and the Hand Gesture dataset, the authors conclude that generative models with Geometry-Aware Convolutional layers produce images of higher quality and greater spatial consistency.

**Strengths:**

The strength of the proposed method is its simplicity. In principle, it is compatible with any convolutional neural network and has a minimal impact on the resulting number of parameters.

**Weaknesses:**

The novelty and empirical evaluation are the main weaknesses of the submission. It is not clear how GeoConv improves on CoordCov and this crucial comparison is missing from the experiment section. The qualitative results presented in the paper are cherry-picked and not convincing enough to prove that the method is useful. Especially for the Hand Gesture GAN and the CelebA conditional VAE experiments, I don't see a substantial difference between a Vanilla-WGAN and a Geo-WGAN in terms of geometrical consistency or image quality. To improve the paper, I suggest including comparisons to CoordConv in all experiments and moving the quantitative evaluation from the appendix to the main text.

The structure of the manuscript is hard to follow. The paragraphs on VAEs and GANs should be moved to the related work section. In addition, a lot of space is dedicated to experiments and details that aren't relevant to the proposed method. For example, the last paragraph of section 3.1.1. contains an analysis of the gender distribution of the generated samples. It is not clear how GeoConv could make the GAN generations more representative. Similarly, the second paragraph describes the center of mass experiment that is not included in the main paper.

**Questions:**

In Figure 5, why is the standard deviation so much greater for Conv2D models compared to GeoConv2D?

How are the random shifts implemented in the GeoConv layer?

Did you use large language models for editing the manuscript?

---

### Official Review · Reviewer_p7s7 · 2023-11-01

**Soundness:** 3 good
**Presentation:** 3 good
**Contribution:** 2 fair
**Rating:** 6
**Confidence:** 3

**Summary:**

The authors improve CoordConv by adding a random shift to the coordinate and combining the two-channel coordinate into one. They test this on image generation task with GANs and VAEs. The new approach seems to show better results compare to CoorConv.

**Strengths:**

- The authors made a compelling motivation on the limitations of CoordConv bias (coordinate bias issue).
- The experiments span a range of scenarios, addressing both face and hand image generation tasks and utilizing two well-known generative models - GAN and VAE.
- Despite its simplicity, the proposed adjustments to CoordConv appears to be effective

**Weaknesses:**

- The paper's novelty appears limited by the straightforward implementation of random shift and channel combination.
- The experiments remain constrained to GANs and VAEs. It would be nice to see the proposed solution apply to other models and tasks, such as U-Net, diffusion models, or tasks beyond image generation. A consistent improvement across these would further highlight the significance of this seemingly simple fix.

**Questions:**

- How were the examples in Figure 4 selected? Were they chosen randomly or cherry-picked?-
- Regarding channel combination, is there an ablation study comparing the speed and result quality to keeping channels separate (but with a random shift)?

Overall my main concern is the limited contributions. My initial rating is borderline leaning slightly toward the positive side mainly due to the potential impact on improving generative model with CNN.